Let’s talk about pain catastrophizing measures: an item content analysis

Crombez Geert geert.crombez@ugent.be 1
De Paepe Annick L. 1
Veirman Elke 1
Eccleston Christopher 2
Verleysen Gregory 3
Van Ryckeghem Dimitri M.L. 1 4 5
1 Deparment of Experimental-Clinical and Health Psychology, Faculty of Psychology and Educational Sciences, Ghent University , Ghent , Belgium
2 Centre for Pain Research, University of Bath , Bath , United Kingdom
3 Ghent University, Research Support Office, Faculty of Psychology and Educational Sciences , Ghent , Belgium
4 Experimental Health Psychology, Faculty of Psychology and Neuroscience, Maastricht University , Maastricht , Netherlands
5 Institute for Health and Behaviour, INSIDE, University of Luxembourg , Esch-sur-Alzette , Luxembourg
Gray Andrew
Electronic publication date: 2020 Mar 4
Publication date: 2020
Volume: 8
Electronic Location ID: e8643
Received 2019 Apr 25; Accepted 2020 Jan 27
Copyright: ©2020 Crombez et al.
Copyright year: 2020
Copyright holder: Crombez et al.
License: This is an open access article distributed under the terms of the Creative Commons Attribution License, which permits unrestricted use, distribution, reproduction and adaptation in any medium and for any purpose provided that it is properly attributed. For attribution, the original author(s), title, publication source (PeerJ) and either DOI or URL of the article must be cited.
License URL: https://creativecommons.org/licenses/by/4.0/

Keywords: Catastrophizing, Worrying, Pain, Validity, Questionnaires, Content validity

Funding: European Union’s Horizon 2020 research Marie Sklodowska-Curie grant 706475 DOLORisk 633491 Dimitri Van Ryckeghem receives funding from the European Union’s Horizon 2020 research and innovation program under the Marie Sklodowska-Curie grant agreement No 706475. Geert Crombez and Elke Veirman have received financial support from DOLORisk, a European Union Horizon 2020 research and innovation programme (Grant 633491) for the research reported in this article. The funders had no role in study design, data collection and analysis, decision to publish, or preparation of the manuscript.

==============================
Background

Concerns have been raised about whether self-report measures of pain catastrophizing reflect the construct as defined in the cognitive-behavioral literature. We investigated the content of these self-report measures; that is, whether items assess the construct ‘pain catastrophizing’ and not other theoretical constructs (i.e., related constructs or pain outcomes) using the discriminant content validity method.

Method

Items (n = 58) of six pain catastrophizing measures were complemented with items (n = 34) from questionnaires measuring pain-related worrying, vigilance, pain severity, distress, and disability. Via an online survey, 94 participants rated to what extent each item was relevant for assessing pain catastrophizing, defined as “to view or present pain or pain-related problems as considerably worse than they actually are” and other relevant constructs (pain-related worrying, vigilance, pain severity, distress, and disability).

Results

Data were analyzed using Bayesian hierarchical models. The results revealed that the items from pain-related worrying, vigilance, pain severity, distress, and disability questionnaires were distinctively related to their respective constructs. This was not observed for the items from the pain catastrophizing questionnaires. The content of the pain catastrophizing measures was equally well, or even better, captured by pain-related worrying or pain-related distress.

Conclusion

Based upon current findings, a recommendation may be to develop a novel pain catastrophizing questionnaire. However, we argue that pain catastrophizing cannot be assessed by self-report questionnaires. Pain catastrophizing requires contextual information, and expert judgment, which cannot be provided by self-report questionnaires. We argue for a person-centered approach, and propose to rename ‘pain catastrophizing’ measures in line with what is better measured: ‘pain-related worrying’.

Introduction

Pain catastrophizing, broadly conceived as an exaggerated mental set brought to bear during painful experiences (Sullivan et al., 2001), has emerged as one of the most important psychosocial predictors of pain, distress, and disability (Edwards et al., 2011; Keefe et al., 2004; Quartana, Campbell & Edwards, 2009). In a meta-analysis of 32 prospective studies, Lewis and colleagues (2015) found that pain catastrophizing was the strongest predictor of persistent pain after total knee arthroplasty. Sobol-Kwapinska and colleagues (2016) reviewed 53 prospective studies investigating various psychosocial risk factors, and found that pain catastrophizing was most strongly linked to acute postsurgical pain. In their review, Fadyl & McPherson (2008) reported preliminary evidence for the role of pain catastrophizing on return to work after injury. In regard to its causal status, studies investigating the processes of change in pain management programs revealed that reductions in pain catastrophizing are key for achieving treatment success (Besen et al., 2017; Burns, Day & Thorn, 2012; Burns et al., 2003; Jensen, Turner & Romano, 2001; Smeets et al., 2006; Turner, Holtzman & Mancl, 2007).

Notwithstanding these achievements, how pain catastrophizing is best measured has not been adequately addressed. In an early commentary, Turner & Aaron (2001) called for an inquiry to determine whether measures of pain catastrophizing really reflect the construct. Despite this call for action, researchers seem to have adopted a pragmatic stance. Comforted by the findings that pain catastrophizing ‘makes a difference’ in predicting pain, distress, and disability (Illari & Russo, 2014), researchers seem to have uncritically embraced the view that pain catastrophizing questionnaires measure pain catastrophizing. We argue that this approach is flawed and deserves scrutiny. First, doubt exists whether the available questionnaires are interchangeable. Indeed, although catastrophizing measures have a good test-retest reliability (e.g.,  Wheeler, Williams & Morley, 2019), correlations between catastrophizing instruments are often only modest (e.g.,  Goubert, Crombez & Danneels, 2005). Second, catastrophizing is a transdiagnostic construct (Gellatly & Beck, 2016; Linton, 2013), and an essential ingredient in many models of psychopathological disorders (Beck, 1979; Clark, 1986; Salkovskis & Clark, 1993). Concerns have been raised about whether catastrophizing as discussed in the pain field is faithful to how it is used in the psychopathology literature (Flink, Boersma & Linton, 2013; Neblett, 2017; Turner & Aaron, 2001). Historically, catastrophizing was introduced by Ellis (1962) and picked up by Beck (1976), who argued that neurosis and emotional disorders are caused by irrational or exaggerated thought patterns, such as catastrophic thinking. Catastrophizing was considered as repeated thinking that a situation is unbearable or terrible when it is just a nuisance (Ellis, 1962), or as a dwelling on the most extreme negative consequences conceivable (Beck, 1976). At the core of these definitions is the idea that a person “views or presents a situation as considerably worse than it actually is” (https://en.oxforddictionaries.com/definition/catastrophize, accessed, 30/09/2016). Third, pain catastrophizing questionnaires should be distinct from questionnaires assessing other theoretical constructs (e.g., pain vigilance, pain-related worry, or fear), and from primary outcomes (e.g., pain severity, pain-related distress, or pain-related disability; (Hirsh et al., 2007). Otherwise, theory building becomes hazardous (Dixon & Johnston, 2019; Goubert, Crombez & Van Damme, 2004; Wideman, Adams & Sullivan, 2009) and the explanatory power of pain catastrophizing may be inflated.

To address these challenges, we investigated to what extent items from pain catastrophizing measures are (a) relevant for the construct ‘pain catastrophizing’ (content validity), and (b) distinct from related constructs (i.e., ‘worrying about pain’, ‘pain vigilance’), and primary outcomes (i.e., ‘pain severity’, ‘pain-related distress’, and ‘pain-related disability’) (discriminant content validity).

Materials & Methods

Participants

Participants were recruited via Prolific (https://www.prolific.ac/), a recruitment application to attract participants for online studies (Peer et al., 2017). Inclusion criteria for participation in the survey were: (1) being 18 years of age or older, (2) having English as a first language, and (3) having access to an internet-enabled device. Furthermore, data from participants were only used for the statistical analyses when participants were able to complete the online assessment in line with given instructions (performance criteria), and took at least 15 min to complete the assessment.

Discriminant Content Validity method

The Discriminant Content Validity (DCV) method provides a quantitative procedure for assessing the content of theory-based measures (for a detailed overview of the methodology, see Bell et al., 2017; Johnston et al., 2014. We describe the DCV in 5 steps.

Step 1: Identification of constructs

Six constructs were identified to be used for the categorization of the items. These constructs were ‘pain catastrophizing’, ‘worrying about pain’, ‘pain vigilance’, ‘pain severity’, ‘pain-related distress’, and ‘pain-related disability’. The constructs ‘worrying about pain’ and ‘pain vigilance’ were selected to investigate to what extent items of pain catastrophizing questionnaires could be clearly differentiated from items of other separate, but theoretically related constructs. The categories ‘pain severity’, ‘pain-related distress’ and ‘pain-related disability’ were selected to investigate to what extent items of pain catastrophizing questionnaires could be clearly differentiated from items of pain outcomes. In the case of insufficient discriminative validity of pain catastrophizing questionnaires or content overlap, research findings about pain catastrophizing may be inflated or confounded. Finally, an ‘other’ category was added to prevent the impression that all items had to be categorized as measures of one of the six predefined constructs. At the same time, the ‘other’ category provided the opportunity to check whether participants understood/followed the given instructions. In particular, we considered it impossible for participants to provide the same extreme scores (i.e., −10 or +10) for an item on all six constructs and the ‘other’ category. In that respect, the ‘other’ category is redundant and scores of the ‘other’ category were not included in the statistical analyses.

Construct definitions

For each of the identified constructs, a definition was formulated. Some of the constructs were, however, considered self-explanatory (e.g., ‘pain severity’, ‘pain-related distress’) and no detailed definition was given. For other constructs, there were multiple definitions available. To avoid the introduction of bias in our findings due to preferring the definition of one theoretical framework over another, we opted to use common language definitions as provided in the Online Oxford Living Dictionaries for English (https://en.oxforddictionaries.com accessed on 30/09/2016). Where necessary, we adapted these definitions to the context of pain. This resulted in the following definitions: (1) pain catastrophizing: ‘To view or present pain or pain-related problems as considerably worse than they actually are’; (2) worrying about pain: ‘To feel troubled or anxious about actual or potential pain or pain-related problems’; (3) pain vigilance: ‘The action or state of keeping careful watch for possible pain’; (4) pain-related distress: ‘Distress related to pain or pain-related problems’; (5) pain-related disability: ‘Being limited in your movements, senses, or activities due to pain’; and (6) pain severity: ‘The intensity or severity of pain’. For the ‘other’ category the following description was provided: ‘The item does not measure any of the previous constructs’.

Selection of pain catastrophizing items

A database was created of all items of available self-report measures assessing pain catastrophizing. Items were included when the questionnaire fulfilled the following criteria: (1) used as part of empirical study indexed in the Web of Science, (2) available in English, and (3) consisting of a (sub-)scale explicitly developed for the assessment of pain catastrophizing in adults. A literature search resulted in 10 potentially relevant questionnaires. We contacted the authors of these questionnaires as well as other key authors (n = 19) in the field of pain to ask for more information on (one of) these questionnaires, and a copy of these questionnaires whenever we did not yet have a copy. In addition, we asked these authors whether they were aware of other pain catastrophizing questionnaires that we might have overlooked. The search for pain catastrophizing instruments was performed during October-November 2016. Of all retrieved questionnaires, some were excluded for the following reasons: (1) no article has been published using the instrument (i.e., the ‘Catastrophizing Thoughts about Pain Scale’, and the ‘Catastrophic Interpretation of Pain Scale’), (2) the instrument was relabeled in a later development phase (i.e., the ‘Catastrophizing Thoughts about Pain Scale’ was relabeled as the ‘Worry About Pain Questionnaire’), (3) no English version of the questionnaire was available (i.e., the ‘Pijn Coping en Cognitie Lijst’), and (4) a standard item format was not used: the ‘Cognitive Errors Questionnaire’ (Lefebvre, 1981) used brief vignettes describing specific situations participants may experience. In addition, the ‘Coping met Pijn Vragenlijst’ was not included because the items were very similar to the ‘Coping Strategies Questionnaires’. Finally, six measures of ‘pain catastrophizing’ were included in the content analysis:

• The catastrophizing subscale (three items, e.g., “When I become aware of my pain, this thought comes through my head: I can’t have a tumour, can I?”) of the Avoidance Endurance Questionnaire (AEQ; Hasenbring, Hallner & Rusu, 2009).

• The catastrophizing subscale (10 items1 , e.g., “I begin to worry that something might be seriously wrong with me”) of the Cognitive Coping Strategies Inventory (CCSI; Butler et al., 1989).

• The catastrophizing subscale (17 items, e.g., “I am disappointed in myself for giving in to the pain”) of the Pain Cognition List (PCL; Vlaeyen et al., 1990).

• The Pain Catastrophizing Scale (13 items, e.g., “When I feel pain, I feel like I can’t go on”; (Sullivan, Bishop & Pivik, 1995).

• The catastrophizing subscale (nine items, e.g., “This pain drives me crazy”) of the Pain-Related Self-Statements Scale (PRSS; Flor, Behle & Birbaumer, 1993).

• The catastrophizing subscale (six items, e.g., “When I feel pain, I feel I can’t stand it anymore”) of the Coping Strategies Questionnaire (CSQ; Rosenstiel & Keefe, 1983).

In order to minimize the task load on participants, items were removed from the final database in case of content overlap. This was the case for five items of the CSQ, which were almost identical to items of the PCS. The final database contained 53 items, of which three were reverse coded items (e.g., PCL: “Thinking too much about the pain only makes it worse”). The set of catastrophizing items used can be found in Table S1.

Selection of items for the other constructs

Items for the contrast constructs ‘worrying about pain’, ‘pain vigilance’, ‘pain severity’, ‘pain-related distress’, and ‘pain-related disability’ were selected from subscales that were considered appropriate for the respective construct. For feasibility reasons, the number of items for each construct was limited. For ‘worrying about pain’, nine items (e.g., “I wonder about the cause of the pain”) were retrieved from the Pain Cognitions Inventory (PCI; Kraaimaat, Bakker & Evers, 1997). For ‘pain vigilance’, ten items (e.g., “I pay close attention to pain”) were retrieved from the vigilance scale of the Pain Vigilance and Awareness Questionnaire (PVAQ; McCracken, 1997). For ‘pain severity’, three items (e.g., “On the average, how severe has your pain been during the last week”) were retrieved from the pain severity subscale of the Multidimensional Pain Inventory (MPI; Kerns, Turk & Rudy, 1985). For ‘pain-related distress’, three items (e.g., “During the past week, how tense or anxious have you been”) were retrieved from the affective distress subscale of the MPI (Kerns, Turk & Rudy, 1985). For ‘pain-related disability’, nine items (e.g., “How much has your pain changed your ability to participate in recreational and other social activities”) were retrieved from the disability subscale of the MPI (Kerns, Turk & Rudy, 1985). This database contained 34 items, of which three items were reverse coded items (e.g., PVAQ: “I find it easy to ignore pain”).

Rating scale of items

Similar to the procedure of Johnston and colleagues (Johnston et al., 2014), participants were instructed to answer two questions per construct for each item. In the first question, participants were instructed to indicate whether—in the context of pain—an item assesses a particular construct (‘yes’ or ‘no’). In the second question, participants rated to what extent they were confident in their judgment using an 11 point scale (0 = 0% confidence to 10 = 100% confidence). For ‘yes’-responses the confidence score remained a positive value, whereas for ‘no’-responses the confidence score was turned into a negative value. As such, outcome scores could range between −10 and +10.

Self-report measures

Participant characteristics

Participants were asked to provide demographic information including date of birth, gender, first language, country of residence, ethnicity, marital status, profession, level of education, and health status (ranging from 1 = excellent health to 5 = poor health).

Graded Chronic Pain Scale

Pain severity of the participants was assessed with the Graded Chronic Pain Scale (GCPS, Von Korff et al., 1992). The GCPS consists of seven items: (1) pain intensity right now; (2) worst pain intensity over the last six months; (3) average pain intensity over the last six months; (4) limitation in daily activities because of pain; (5) limitation in recreational, social, and family activities within the last six months; (6) limitation in the ability to work because of pain within the last six months; and (7) the number of days disabled within the last six months. All items, except for the number of days disabled, are scored on an 11-point scale (0–10). Following guidelines provided by Von Korff and colleagues (1992), participants were categorized into one of five grades: grade 0 (‘pain free’), grade 1 (‘low disability-low intensity’), grade 2 (‘low disability-high intensity’), grade 3 (‘high disability-low intensity’), and grade 4 (‘high disability-high intensity’). Research indicated that this questionnaire is reliable and valid for assessing pain and disability in general populations (Von Korff et al., 1992; Goubert, Crombez & De Bourdeaudhuij, 2004; Dixon, Pollard & Johnston, 2007).

Detection of careless responding

Two strategies were used to detect careless responding (Meade & Craig, 2012). First, three items from the Instructional Manipulation Check (IMC, e.g., “It’s important that you pay attention to this study. Please tick ’yes’ and ’30%’ for all definitions”) were intermixed with the DCV items (see also Oppenheimer, Meyvis & Davidenko, 2009). Second, participants were asked to indicate their level of attention during the online survey (1 = almost no attention; 2 = very little attention; 3 = some of my attention; 4 = most of my attention; 5 = my full attention), and whether, in their honest opinion, we should use their data (“yes or no”; Meade & Craig, 2012).

Procedure

The study was performed in line with the ethical guidelines of the Ethics Review Panel of the University of Luxembourg. The online survey was constructed using LimeSurvey 2.00 and a link to the survey was distributed via Prolific Academic. After opening the Web page and formally giving informed consent (online), participants were provided with the instructions of the DCV method, and two non-related examples on how the DCV can be completed. Participants were provided with one of three item sets. The final set of 87 items (excluding IMC items) was split into three item sets because piloting revealed that participants became fatigued when they attempted to categorize too many items. Each item set contained a random selection of items (stratified randomization on questionnaire) and one shared item drawn from each questionnaire (version 1: 25 unique + 11 shared items; version 2: 25 unique + 11 shared items; version 3: 26 unique + 11 shared items). In addition, three IMC items were added to each version. The order of item presentation was random for each participant. Also, the order of presentation of the constructs, listed above each question, differed between participants (three possible orders which remained consistent throughout a person’s assessment). After participants completed the DCV questions, they gave their demographic information, and answered questions assessing their pain (GCPS) and questions to detect careless responding. The online assessment lasted a mean of 35.4 min (SD = 15.8). Participants were rewarded two English pounds for participation.

Statistical analysis

Results were analyzed using Bayesian hierarchical models (JAGS version 4.3.0) in R version 3.6.0 (R Core Team, 2019) to ensure that estimates did not fall outside the actual response range [−10 to 10]. In the model a different mu parameter was estimated for each construct or measure, depending on the research question (see below). In addition, a random effect for subject and item was added. All parameters received vague priors (normal distributions with a very large standard deviation) in order to let the data speak for itself. The dependent variable was the DCV outcome score (ranging from −10 to 10). The mu parameters come from a truncated normal distribution [−10, 10] so the credibility intervals only contain sensible values. To generate the posterior samples, we used 4 chains with 20,000 iterations each, 5,000 being discarded as burn in. Traceplots and Rhat values of 1 indicated that all the chains for the mu parameters reached convergence. Because participants reported difficulties in scoring the reverse coded items, these items were excluded from all analyses.

First, as a manipulation check we investigated whether the items from questionnaires assessing ‘pain catastrophizing’, ‘worrying about pain, ‘pain vigilance’, ‘pain severity’, ‘pain-related distress’, and ‘pain-related disability’, were indeed most relevant for measuring their respective construct. Separate analyses were run for each measure. A Bayesian hierarchical model was fitted with construct as a fixed effect and subject and item as random effects.

Second, to identify which catastrophizing questionnaires scored highest on ‘pain catastrophizing’, we investigated the effect of measure (AEQ, CSQ, CSSI, PCL, PCS, and PRSS) on the outcome scores for the construct ‘pain catastrophizing’ only. A Bayesian hierarchical model was fitted with measure as a fixed effect and subject and item as random effects.

Third, we investigated to what extent items of each pain catastrophizing measure were rated to be distinctively associated with the construct of pain catastrophizing, and less to the other constructs. For each pain catastrophizing questionnaire a separate Bayesian hierarchical model was fitted, comparing the score for the construct ‘pain catastrophizing’ to those for the five other constructs (‘worrying about pain’, ‘pain vigilance’, ‘pain severity’, ‘pain-related distress’, and ‘pain-related disability’). The models contained construct as a fixed effect and subject and item as random effects.

Finally, a separate Bayesian hierarchical model was fitted for each item of the pain catastrophizing questionnaires. The models included construct as a fixed effect and subject as a random effect. For all models, significance was evaluated at the 5% significance level (two-sided). Estimated mu parameters (μ ˆ) and their associated 95% credibility intervals (CI) are reported.

Results

Participants

Data from 138 participants were collected. After application of the inclusion criteria (see ‘Participants’), 44 participants were excluded from further analyses. More specifically, one participant failed to respond correctly on the IMC items, 36 participants provided unreliable data (i.e., at least one item was scored as −10 or +10 for all constructs), two participants stated that English was not their first language, four participants completed the questionnaire in less than 15 min, and one participant indicated at the end of the questionnaire that his/her data should not be used. The final sample contained 94 participants (mean age of 36 years, SD = 12; 59 males, 63%). Most participants reported their ethnicity as Caucasian (n = 87, 93%). Many participants were single (n = 39, 41%) or living with a partner (n = 41, 44%). The majority were from the United Kingdom (n = 55, 59%) followed by the United States (n = 28, 30%). Sixty-nine participants followed or were following a university (college) program (73%).

The large majority of participants (n = 76, 81%) were in at least good health (excellent health: n = 14 (15%); very good health: n = 34 (36%); good health: n = 28 (30%); fair health: n = 12 (13%); poor health: n = 6 (6%)). On the GCPS, participants’ mean current pain was 2.22 (SD = 2.20; median = 2, IQR = 3). Participants’ mean worst pain level during the past six months and mean average pain level were 5.05 (SD = 2.99; median = 5, IQR = 5) and 3.48 (SD = 2.67; median = 3, IQR = 5), respectively. Furthermore, participants reported a mean pain interference with daily activities of 2.98 (SD = 2.98; median = 2, IQR = 5.75), a mean change of their ability to take part in recreational, social and family activities of 2.80 (SD = 2.95; median = 2, IQR = 5), and a mean change of their ability to work of 2.55 (SD = 2.92; median = 1.5, IQR = 5). Finally, participants reported a mean of 15 days off normal task within the last six months (SD = 32; median = 3, IQR = 10). Calculation of the pain grade showed that few participants were classified in grade 0 (n = 8, 8%). Most participants were classified in grade 1 (n = 56, 60%). Six participants were classified in grade 2 (6%), 13 participants in grade 3 (14%), and 11 participants in grade 4 (12%).

Content validity of alternative constructs

As a manipulation check, we investigated whether the items from questionnaires assessing ‘worrying about pain’ (PCI), ‘pain vigilance’ (PVAQ), ‘pain severity’ (MPI - pain severity subscale), ‘pain-related distress’ (MPI - affective distress subscale), and ‘pain-related disability’ (MPI - disability subscale) were indeed most relevant for measuring their respective construct. Findings are displayed in Fig. 1. Overall, participants have performed the DCV method adequately. Almost all measures loaded highest and distinctively on their respective construct. There was one exception, i.e., participants were unable to differentiate between items of ‘worrying about pain’ and items of ‘pain-related distress’. More detailed results are given in File S1.

Figure 1 Estimates and associated 95% credibility intervals (CI) of the relevance score for the PCI (worry about pain), PVAQ (pain vigilance) and MPI (affective distress; disability; pain severity) on each construct.

The estimate and CI for the construct each measure is designed to assess is indicated in black. The estimate and CI for the other constructs is depicted in grey. Included scales were (subscales of) the (A) PCI (Pain Cognitions Inventory), (B) PVAQ (Pain Vigilance and Awareness Questionnaire), (C) MPI - affective distress (Multidimensional Pain Inventory), (D) MPI - disability (Multidimensional Pain Inventory). (E) MPI - pain severity (Multidimensional Pain Inventory).

Content analyses of pain catastrophizing measures

Content analyses between pain catastrophizing measures

To identify which catastrophizing measures scored highest on ‘pain catastrophizing’, we investigated the effect of measure (AEQ, CSQ, CSSI, PCL, PCS and PRSS) on the outcome scores for the construct ‘pain catastrophizing’ only. The estimated mu parameters and the 95% credibility intervals for each measure on ‘pain catastrophizing’ are displayed in Fig. 2. The CSQ had the highest score on the construct ‘pain catastrophizing’. The AEQ had the second highest score and did not score significantly lower than the CSQ (Δ = 0.13, 95% CI [−1.25 to 1.52]). The CCSI, the PRSS, the PCS, and the PCL scored significantly lower than the AEQ and the CSQ. The items of the AEQ, CSQ, CCSI, PCS, and PRSS scored on average significantly higher than 0 on the construct ‘pain catastrophizing’. The items of the PCL scored significantly lower than 0, indicating that participants believed that the items did not measure ‘pain catastrophizing’.

Figure 2 Estimates and associated 95% credibility intervals of the relevance score for each of the six pain catastrophizing measures on pain catastrophizing.

The six pain catastrophizing measures were (subscales of) the AEQ (Avoidance Endurance Questionnaire), CSQ (Coping Strategies Questionnaire), CCSI (Cognitive Coping Strategies Inventory), PCS (Pain Catastrophizing Scale), PCL (Pain Cognition List), and PRSS (Pain-Related Self-Statements Scale).

In summary, there is variability in the extent to which instruments measure ‘pain catastrophizing’. The AEQ (Hasenbring, Hallner & Rusu, 2009) and the CSQ (Rosenstiel & Keefe, 1983) have the highest ratings for the ‘pain catastrophizing’ construct. Participants did not endorse the PCL (Vlaeyen et al., 1990) as measuring ‘pain catastrophizing’.

Content analyses per pain catastrophizing measure

In this section, we investigated to what extent items of each pain catastrophizing measure were rated to be distinctively associated with the construct ‘pain catastrophizing’, and less to the other constructs. For each catastrophizing questionnaire, a different model was fit, comparing the score for the construct ‘pain catastrophizing’ to those for the five other constructs (‘worrying about pain’, ‘pain vigilance’, ‘pain severity’, ‘pain-related disability’, and ‘pain-related distress’). Figure 3 displays the estimated mu parameters for each construct and their 95% credibility intervals per measure of pain catastrophizing.

Figure 3 Estimates and associated 95% credibility intervals of the relevance score for each of the six pain catastrophizing measures on each construct.

The estimate and CI for the construct each measure is designed to assess is indicated in black. The estimate and CI for the other constructs is depicted in grey. The six pain catastrophizing measures were (subscales of) the (A) AEQ (Avoidance Endurance Questionnaire), (B) CSQ (Coping Strategies Questionnaire), (C) CCSI (Cognitive Coping Strategies Inventory), (D) PCL (Pain Cognition List), (E) PCS (Pain Catastrophizing Scale), and (F) PRSS (Pain-Related Self-Statements Scale).

AEQ. The items of this measure scored significantly higher on ‘pain catastrophizing’ than on ‘pain-related disability’ (Δ = 10.41, 95% CI [9.06–11.78]), ‘pain severity’ (Δ = 8.64, 95% CI [7.28–10.00]), and ‘pain vigilance’ (Δ = 6.65, 95% CI [5.30–8.01]). However, the items scored significantly lower on ‘pain catastrophizing’ than on ‘worrying about pain’ (Δ = −1.68, 95% CI [−3.01 to −0.32]). There was no significant difference with the score on ‘pain-related distress’ (Δ = 0.58, 95% CI [−0.78–1.94]).

CSQ. The items of this measure scored significantly higher on ‘pain catastrophizing’ than on ‘pain-related disability’ (Δ = 6.25, 95% CI [5.14–7.37]), ‘pain severity’ (Δ = 2.29, 95% CI [1.19–3.41]), and ‘pain vigilance’ (Δ = 8.23, 95% CI [7.12–9.34]). Yet, the items scored significantly lower on ‘pain catastrophizing’ than on ‘pain-related distress’ (Δ = −2.93, 95% CI [−4.04 to −1.82]). There was no significant difference with the score on ‘worrying about pain’ (Δ = 0.30, 95% CI [−0.81–1.41]).

CCSI. The items of this measure scored significantly higher on ‘pain catastrophizing’ than on ‘pain-related disability’ (Δ = 8.27, 95% CI [7.25–9.30]), ‘pain severity’ (Δ = 4.33, 95% CI [3.30–5.37]), and ‘pain vigilance’ (Δ = 5.03, 95% CI [4.00–6.07]). However, the items scored significantly lower on ‘pain catastrophizing’ than on ‘worrying about pain’ (Δ = −1.12, 95% CI [−2.15 to −0.09]). There was no significant difference with the score on ‘pain-related distress’ (Δ = −0.81, 95% CI [−1.86–0.22]).

PRSS. The items of this measure scored significantly higher on ‘pain catastrophizing’ than on ‘pain-related disability’ (Δ = 4.66, 95% CI [3.61–5.71]) and ‘pain vigilance’ (Δ = 5.20, 95% CI [4.14–6.25]). However, the items scored significantly lower on ‘pain catastrophizing’ than on ‘pain-related distress’ (Δ = −1.91, 95% CI [−2.96 to −0.86]). There were no significant differences with the scores on ‘pain severity’ (Δ = −0.90, 95% CI [−1.94–0.14]) and ‘worrying about pain’ (Δ = 0.10, 95% CI [−0.95–1.15]).

PCS. The items of this measure scored significantly higher on ‘pain catastrophizing’ than on ‘pain-related disability’ (Δ = 5.12, 95% CI [4.28–5.98]) and ‘pain vigilance’ (Δ = 3.77, 95% CI [2.92–4.64]). However, the items scored significantly lower on ‘pain catastrophizing’ than on ‘pain-related distress’ (Δ = −4.74, 95% CI [−5.60 to −3.89]) and ‘worrying about pain’ (Δ = −3.90, 95% CI [−4.76 to −3.05]). There was no significant differences with the score on ‘pain severity’ (Δ = 0.82, 95% CI [−0.03–1.67]).

PCL. The items of this measure scored significantly higher on ‘pain catastrophizing’ than on ‘pain-related disability’ (Δ = 1.55, 95% CI [0.73–2.38]), albeit that the score on ‘pain catastrophizing’ was negative. Moreover, the items scored significantly lower on ‘pain catastrophizing’ than on ‘pain-related distress’ (Δ = −3.87, 95% CI [−4.68 to −3.04]) and ‘worrying about pain’ (Δ = −3.91, 95% CI [−4.73 to −3.09]). There were no significant differences with the scores on ‘pain severity’ (Δ = −0.06, 95% CI [−0.87 to 0.75]) and ‘pain vigilance’ (Δ = −0.58, 95% CI [−1.40–0.24]).

In summary, none of the six instruments of pain catastrophizing distinctively assessed ‘pain catastrophizing’. Most instruments have content that was equally well, or even better captured by the constructs ‘worrying about pain’ or ‘pain-related distress’.

The five highest and lowest scoring pain catastrophizing items

The results of the five items with the highest and lowest scores on the construct ‘pain catastrophizing’ are reported here. A complete list detailing the performance of each item per construct is presented in Table S1. A separate model was fitted for each of the items. Results for the five highest and lowest scoring pain catastrophizing items are summarized in Table 1. The five highest scoring items had good positive mean item scores (>5.00), indicating that these items were endorsed to measure ‘pain catastrophizing’. However, these items did not distinctively measure ‘pain catastrophizing’. These items were also found to load equally well, or even better on ‘worrying about pain’ or ‘pain-related distress’. The five lowest scoring items had negative item scores, indicating that these items were considered not to measure ‘pain catastrophizing’.

Table 1 The five highest and lowest scoring pain catastrophizing items across pain catastrophizing measures.

	Item	Measure	Pain catastrophizing
μ ˆ[CI]	Pain-related disability
μ ˆ[CI]	Pain-related distress
μ ˆ[CI]	Pain severity μ ˆ[CI]	Pain vigilance
μ ˆ[CI]	Worrying about pain
μ ˆ[CI]	
Best items	When I’m in pain, it’s terrible and I think it’s never going to get any better”	PCS/CSQ	7.41
[5.36 to 9.44]	−3.56*
[−6.39 to −0.75]	7.21
[5.41 to 9.00]	4.70
[2.14 to 7.26]	−1.62*
[−4.55 to 1.31]	6.21
[3.76 to 8.65]	
	This will never end	PRSS	6.58
[4.51 to 8.64]	−4.61*
[−7.23 to −2.00]	5.74
[3.23 to 8.24]	−0.75*
[−3.88 to 2.36]	−4.58*
[−7.28 to −1.90]	4.75
[1.91 to 7.59]	
	I can’t go on anymore	PRSS	5.92
[3.60 to 8.24]	2.58
[−0.33 to 5.48]	8.05*
[6.33 to 9.76]	3.54
[0.66 to 6.41]	−5.48*
[−7.45 to −3.52]	3.63
[0.88 to 6.38]	
	When I feel pain, I feel my life isn’t worth living	CSQ	5.90
[4.64 to 7.15]	−2.21*
[−3.73 to −0.67]	8.28*
[7.58 to 8.99]	0.54*
[−1.11 to 2.20]	−4.50*
[−5.80 to −3.17]	3.89*
[2.35 to 5.43]	
	When I’m in pain, I feel I can’t stand it anymore	PCS/CSQ	5.49
[3.37 to 7.62]	−0.49*
[−3.35 to 2.34]	6.82
[5.15 to 8.46]	5.32
[2.99 to 7.64]	−5.00*
[−7.11 to −2.88]	1.88*
[−0.89 to 4.66]	
Worst items	I need to take some pain medication	PRSS	−7.06 [−8.68 to −5.43]	−3.57*
[−6.49 to −0.68]	−1.71*
[−4.73 to 1.30]	2.97*
[−0.02 to 5.96]	−0.20*
[−3.43 to 3.01]	−3.09
[−5.96 to −0.20]	
	When I’m in pain, I anxiously want the pain to go away	PCS	−6.54
[−8.36 to −4.69]	−6.92
[−8.85 to −4.99]	5.61*
[3.03 to 8.18]	−3.74*
[−6.72 to −0.77]	−1.68*
[−4.79 to 1.41]	6.27*
[3.84 to 8.71]	
	I make sure I protect myself from extra pain	PCL	−5.55
[−6.64 to −4.46]	−5.40
[−6.58 to −4.21]	−2.46*
[3.98 to −0.94]	−2.10*
[−3.69 to −0.52]	6.91*
[5.81 to 8.04]	1.18*
[−0.41 to 2.77]	
	I think I am always very tense	PCL	−5.51
[−7.48 to −3.54]	−6.04
[−7.79 to −4.30]	−0.17*
[−2.96 to 2.60]	−5.93
[−7.83 to −4.02]	−3.37*
[−5.95 to −0.79]	1.31*
[−1.33 to 3.94]	
	I am disappointed in myself for giving in to the pain	PCL	−4.88
[−7.33 to −2.41]	−4.64
[−7.16 to −2.15]	4.57*
[1.79 to 7.35]	−4.57
[−7.33 to −1.78]	−5.88
[−8.25 to −3.53]	0.59*
[−2.47 to 3.66]	
Notes.

mu estimated mean in the population

CI 95% credibility interval

* p < 0.05, indicating a significant difference in score with the construct ‘pain catastrophizing’.

In summary, none of the five highest scoring items of the pain catastrophizing instruments distinctively assessed ‘pain catastrophizing’. They have content that was equally well, or even better, captured by the constructs ‘worrying about pain’ or ’pain-related distress’. The five lowest scoring items did not seem to have any content related to ‘pain catastrophizing’.

Additional analyses - influence of item set group and moderators

In a series of additional analyses, we checked whether participants from the three item set groups were comparable with respect to their scores. Item set group did not moderate the outcome score for the shared items on the different constructs. These groups did not differ with respect to potentially moderating variables (pain group, age, and gender). Therefore, item set group was not considered as a fixed effect in the analyses.

Second, we assessed whether pain group, age, and gender could moderate the relationship between the constructs and the score allocated to the items. The overall pattern of results did not differ as a function of pain group, age, and gender. Detailed results of these additional analyses are available in File S2.

Discussion

This study investigated the content of pain catastrophizing measures. Participants rated the extent to which each item was relevant for measuring ‘pain catastrophizing’, other theoretically related constructs (‘worrying about pain’, ‘pain vigilance’), and pain outcome constructs (‘pain severity’, ‘pain-related distress’, and ‘pain-related disability’). The results can be readily summarized. First, there was variability in the extent to which instruments measure ‘pain catastrophizing’. The item content of the AEQ (Hasenbring, Hallner & Rusu, 2009) and the CSQ (Rosenstiel & Keefe, 1983) best reflected ‘pain catastrophizing’. Second, despite some instruments measure ‘pain catastrophizing’ better than others, none of the instruments purporting to measure pain catastrophizing distinctively assessed ‘pain catastrophizing’. Most instruments had content that was equally well, or was even better, captured by the constructs of ‘worrying about pain’ or ‘pain-related distress’. Third, this pattern was robust. It did not substantially differ between individuals reporting disabling pain, and it was also observed for the five items assigned the highest values for ‘pain catastrophizing’. Current findings confirm the doubts raised about whether pain catastrophizing measures actually assess ‘pain catastrophizing’ as defined in the cognitive-behavioral literature (Eccleston & Crombez, 2017; Eccleston et al., 2012; Turner & Aaron, 2001).

To our knowledge, this study is the first to empirically investigate the content of pain catastrophizing instruments, addressing the extent to which items are relevant for the construct ‘pain catastrophizing’ and to what extent these items are distinct from other related constructs and pain outcome constructs. Despite being an essential property of an instrument (Terwee et al., 2007), content validity is often neglected and overlooked at the expense of other forms of validity such as construct and criterion validity (Dixon & Johnston, 2019; Wiering, Boer & Delnoij, 2017). Furthermore, content validity is often confused with face validity (i.e., the extent to which an instrument appears to be valid), which is technically not a form of validity (Lilienfeld et al., 2017). With the advent of patient-reported outcomes, various guidelines for designing new instruments are available and increasingly used (Magasi et al., 2012). These provide guidance for ensuring and assessing content validity. Here, we addressed the content validity of already available instruments. Such an approach is helpful in identifying and highlighting measurement and conceptual problems with established instruments, which are too often taken for granted to be valid (Crombez, 2015; Fried, 2017; Grossman, 2011; Lauwerier et al., 2015). Adopting the Oxford Living Dictionary definition, we defined ‘pain catastrophizing’ as “to view or present pain or pain-related problems as considerably worse than they actually are”. This definition is in line with how the construct is used in psychopathology. Furthermore, many studies on pain catastrophizing seem to adhere to this meaning. For example, Sobol-Kwapinska and colleagues (2016) discussed pain catastrophizing as exaggerating the negative aspects of a situation. Similarly, in their critical review of pain catastrophizing, Quartana and colleagues (2009) defined pain-related catastrophizing as “a set of exaggerated and negative cognitive and emotional schema brought to bear during actual or anticipated painful stimulation” (p. 2).

The results of current content analysis are sobering. None of the six instruments distinctively measured ‘pain catastrophizing’. Similar analyses at the item level corroborated this finding. Further, instruments purporting to measure ‘pain catastrophizing’ were considered to perform equally well, or even better in their ability to assess ‘worrying about pain’. This may not come as a surprise. Many authors consider pain catastrophizing as an extreme instance of worrying (Davey & Levy, 1998; Harvey & Greenall, 2003). It then makes sense that worrying and catastrophizing are strongly intertwined. An identical pattern was found for ‘pain catastrophizing’ and ‘pain-related distress’. Such a strong overlap was not expected. Theoretically, pain catastrophizing is a precursor of pain-related distress (Rode, Salkovskis & Jack, 2001; Vlaeyen & Linton, 2000). We anticipated that cause and effect would be distinguishable. This was not observed. Reconsidering the literature, Beck (1976) and Ellis (1962) identified catastrophizing as part of clinical forms of distress and anxiety. Also, Hirsch and colleagues (2007) came to the same conclusion after showing that pain catastrophizing did offer little predictive value beyond negative mood.

We recognize that our conclusion that measures of pain catastrophizing actually do not measure ‘pain catastrophizing’, as understood by our participants, is a major challenge to models that hold pain catastrophizing as a core concept. We were, however, careful to ensure that our results are internally valid. We checked whether participants correctly performed the task. Also, we showed that our method for assessing content validity worked for instruments measuring other constructs (i.e., ‘worrying about pain’, ‘vigilance’, and ‘pain severity’). In contrast to the items from the pain catastrophizing measures, the items of these instruments had fidelity. Only ‘pain-related distress’ and ‘pain-related worrying’ were strongly interrelated. We also investigated whether different results emerged as a function of age, gender, and pain status. Overall, the pattern of results remained the same. The few significant moderation effects observed, were not compelling and did not lead to alternative interpretations.

A natural response to our results may be to embark on developing a novel questionnaire. This seems appropriate at first sight. Our database of items may be a good starting point, and new items that comprehensively reflect facets of ‘pain catastrophizing’ may be designed. Also, guidelines for instrument development are available (Bossuyt et al., 2015; DeWalt et al., 2007). However, we argue that pain catastrophizing, defined as “to view or present pain or pain-related problems as considerably worse than they actually are”, cannot be assessed by self-report measures. At the heart is a measurement issue common in psychology. Instruments with discriminative ambitions (Kirshner & Guyatt, 1985), such as pain catastrophizing questionnaires, need a reference standard to substantiate their diagnostic test accuracy (Bossuyt et al., 2015). For catastrophizing, this means establishing that someone’s belief about pain or pain-related problems is incorrect or exaggerated. To judge such an error, one needs three things: an objective measure of the actuality (‘how bad things really are’); the population standard for worrying about pain (‘how bad everyone else think they are’); and an expert judgment that the individual’s perception crosses the threshold into extreme (‘it is not as bad as they believe’). Even when one can measure the extent of a real catastrophe, and if population standards for worrying about pain were established, somebody will always have to decide whether the experience of a person is worse than the situation demands. Such decisions require contextual information, which self-report instruments do not provide. In summary, we do not think it is possible to measure pain catastrophizing using self-report questionnaires. Therefore, we propose to adopt a person-centered approach, and to rename ‘pain catastrophizing’ measures in line with what is better measured: ‘pain-related worrying’ (Eccleston & Crombez, 2017; Eccleston et al., 2012). Such renaming may also have clinical implications. It may invite clinicians to explore what patients are worried about. In contrast, the label ‘pain catastrophizing’ may easily elicit inappropriate referral to mental health professionals with expertise in abnormal and extreme cognition and affect, leaving patients feeling less understood and more stigmatized (Amtmann et al., 2018; De Ruddere & Craig, 2016).

We do not want to throw the baby out with the bathwater. Instruments purporting to measure pain catastrophizing have proven to be empirically useful in predicting and explaining a range of outcomes. Indeed, a change in name does not change the successes that have been achieved with these questionnaires. To improve the theory that emerges from this successful empiricism we suggest the following. First, we propose to always look beyond the aggregate scores of instruments, and to unpack instruments to their basic constituents (Eccleston & Crombez, 2017). That way, we may find (dis)similarities with other constructs, and identify the critical components in an explanatory model of pain, distress, and disability. Some of these components may relate to repetitive negative thinking (Davey & Levy, 1998; Flink, Boersma & Linton, 2013), appraisal (Sullivan et al., 2001), expectancies about future pain and disability (Peerdeman et al., 2016), cognitive intrusion (Attridge et al., 2015), or personal inadequacies (Davey & Levy, 1998). Second, if researchers truly want to investigate the causal role of psychological variables in predicting pain, pain-related distress and disability, we need to carefully check and appropriately control for confounding factors. Here, we found content overlap with relevant pain outcomes, in particular pain-related distress. We recommend that in future studies on the role of psychosocial variables, content overlap with outcomes is systematically addressed. The DCV method used here can be easily adopted (Dixon & Johnston, 2019).

Our findings are limited by the methods we chose. First, the content validity was investigated using people willing and able to participate in an online study. No experts or patients were involved. Second, we opted for a quantitative analysis of content validity. Other methods are possible, and may provide insight about how exactly participants understand items (Amtmann et al., 2018). One promising procedure, for example, is cognitive interviewing (Beatty & Willis, 2007; Willis, 2015). Third, we adopted the DCV method (Johnston et al., 2014), but our statistical analyses differed. Their research with the DCV method involves a low number of participants, and uses primarily one-sample or paired t-tests. Hence, their results strongly depend on sample size and statistical power. Our approach for content validity involves analyses using Bayesian hierarchical models, which requires a larger number of participants. Notwithstanding this, it has several advantages (e.g., combined analyses on item and measure level, group and construct comparisons, and credibility intervals within actual possible range). Fourth, our definitions of constructs were adapted from the Oxford Living Dictionary, which may not always concur with scientific definitions (Dixon, Pollard & Johnston, 2007). However, we believe that this was not a problem for pain catastrophizing, as this definition was in line with the meaning of the term in the cognitive-behavioral literature on psychopathology. Fifth, the content overlap between pain catastrophizing and pain-related distress deserves further scrutiny. We did not provide a precise definition of pain-related distress, as we believed this term to be self-explanatory. However, this choice may have left the construct ill-defined and overinclusive. For that reason we prefer, as yet, the term pain-related worrying instead of pain-related distress as an alternative term for ‘pain catastrophizing’. Sixth, we have not included all pain catastrophizing instruments. Most notably, we excluded the CEQ, which had a different item format from the standard catastrophizing questionnaires. Seventh, we only addressed the content validity of catastrophizing measures, in particular the relevance of items for measuring pain catastrophizing. We did not address their construct validity (‘The degree to which the instrument scores are consistent with theory and hypotheses’) and their criterion validity (‘The degree to which the instrument scores are related to an outcome/criterion’) (Mokkink et al., 2010). Eighth, we recognize that the measurement challenges we describe are not unique to pain catastrophizing. In pain research, this means that the measurement of common constructs such as ‘somatization’, ‘kinesiophobia’, and ‘hypervigilance’ will have problems inasmuch as they rely on an objective measure of the actual, a population normative standard, and an expert judgment that the person’s perception is extreme. If assessed through decontextualized self-report, they will all be found wanting. Greater scrutiny of these approaches and discussion of alternatives is needed (Cella et al., 2010; Crombez, 2015; Crombez et al., 2009; Grossman, 2011). So, let’s talk…

Conclusions

Pain catastrophizing, defined as “to view or present pain or pain-related problems as considerably worse than they actually are” is not measured by current self-report questionnaires of pain catastrophizing. It is unlikely that it ever will. The construct ‘pain catastrophizing’ requires contextual information, and expert judgment, which cannot be provided by self-report questionnaires. We argue for a person-centered approach, and propose to rename ‘pain catastrophizing’ measures in line with what is better measured: ‘pain-related worrying’.

Supplemental Information

Table S1 The estimates and their associated 95% credibility interval [CI] per construct for each item of the six pain catastrophizing instruments

∗p < 0.05, indicating that the score of an item for that particular construct is significantly different from the score for the construct ’Pain catastrophizing.’

Click here for additional data file.

File S1 Content validity of alternative constructs

Click here for additional data file.

File S2 Additional analyses

Click here for additional data file.

We thank Jana Bellaert, Lisa Krämer, Caroline Bell, Madeline Gross, Eva Diana Franz and Marina Spezzacatena for their help in the study.

Additional Information and Declarations

Competing Interests

Author Contributions

Human Ethics

Data Availability

1 Two items (i.e., “I imagine the pain becoming even more intense and hurtful” and “I tell myself that I don’t think I can bear the pain any longer”) of the catastrophizing subscale of the CCSI were erroneously not included in the content analysis

The authors declare there are no competing interests.

Geert Crombez and Christopher Eccleston conceived and designed the experiments, authored or reviewed drafts of the paper, and approved the final draft.

Annick L. De Paepe conceived and designed the experiments, analyzed the data, prepared figures and/or tables, authored or reviewed drafts of the paper, and approved the final draft.

Elke Veirman conceived and designed the experiments, prepared figures and/or tables, authored or reviewed drafts of the paper, and approved the final draft.

Gregory Verleysen analyzed the data, authored or reviewed drafts of the paper, and approved the final draft.

Dimitri M.L. Van Ryckeghem conceived and designed the experiments, performed the experiments, authored or reviewed drafts of the paper, and approved the final draft.

The following information was supplied relating to ethical approvals (i.e., approving body and any reference numbers):

The study was performed according to the ethical guidelines of the Ethics Review Panel of the University of Luxembourg.

The following information was supplied regarding data availability:

Data is available at OSF: Crombez, Geert, Annick De Paepe, Elke Veirman, and Van ryckeghem Dimitri. 2020. “Let’s Talk about Pain Catastrophizing Measures: An Item Content Analysis.” OSF. January 6. doi 10.17605/OSF.IO/4KHYQ.

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
