# Peer review of "Let’s talk about pain catastrophizing measures: an item content analysis"

_PeerJ, doi:10.7717/peerj.8643_

## Round 0.1 · original submission · Major Revisions

This is potentially an important manuscript and the reviewers have jointly made several positive comments about the value of your work. They have, however, also raised some issues that need addressing and some will require very careful consideration on your part. Please respond to each of their comments, and my additional comments below, with a point-by-point response, explaining in each case what was changed in the manuscript or why no changes were made in response to that particular point.

As a biostatistician, I have some concerns about parts of the statistical analyse (e.g. the comparisons between the apparently randomised groups, which requires specific justification) and about the model selection process. It’s possible that these concerns are arising from details being omitted from what is already a rather long manuscript, or from me not picking up points as you were expecting readers to do. Either way, there is a need to make the analysis approach clearer and to justify the statistical decisions made. Reviewer #1 makes some comments on this point also.

Given the length of the manuscript, you might want to move some of the results to a supplement (or supplements). The manipulation checks (25 lines) would be a candidate for this (if this is retained beyond a descriptive presentation), and you could consider whether the alternative constructs (3.2.2) section (55 lines) could be summarised in the manuscript with the full version as a supplement. I’ll leave these decisions entirely up to you, but I think that anything you can do to focus the reader’s attention during the lengthy results would be beneficial.

Another important point that requires more attention here is that lack of evidence is not equal to evidence of lack. You are consistent in correctly presenting confidence intervals, but there is a lack of interpretation of these intervals. This is one of the side-effects of null hypothesis significance testing and while I have above noted the length and suggested a shortening of the manuscript’s results, I will also ask if you can include the CIs more in your interpretation in the discussion. I feel that forest plots could be a useful way for you to present a great deal of information in a relatively succinct format here. I also suggest more caution and qualifying, even if it adds slightly to the word count, when presenting your findings in the discussion. Reviewer #2 notes some specific concerns they had about some findings being overstated.

There is also a need for some discussion of model diagnostics and a consideration of the issue of multiplicity here.

As well as some helpful corrections, Reviewer #1 raises some excellent points about defining pain catastrophizing. They also note that content validity is a broad term and I absolutely agree with their point that the distinction between content validity and discriminant content validity warrants care. I think that all of their other comments are also important for you to address.

Reviewer #2 raises some equally if not more challenging questions. Their relatively minor comments about the pain groups and the use of the instruments would address some questions that I’m sure readers would wonder about. Their points about how distress is defined are challenging and need careful attention, as do their points about the history of the term “pain catastrophizing”. Their other comments, including about the subscales of the PCS, are also all worthy of careful response.

My specific comments:

Line 23: I don’t think the “of” after “comprising” is necessary, although you could say “consisting of” instead.

Line 25: Perhaps “Using” rather than “Via” here?

Line 31: Do you want a comma after “better”?

Line 47: There seems be an issue with the formatting at the start of the line.

Lines 65–67: I think this argument would be stronger if the instruments’ reliability were first, or even subsequently, established; then a lack of correlation would indicate something about their content. Without this step, the poor correlation could simply indicate poor reliability.

Line 79: Just “distinct” would work here I think. If you agree, you might also want to look at its inflected forms, although some of those I would leave as is despite this suggestion.

Lines 86–87: This sentence seems like methods to me. You could also consider starting the following sentence as “We aimed to investigate…” to clearly signal the study aim at this stage.

Line 96–98: The 4th and 5th criteria here sound more like data cleaning/validation steps than determining whether someone was eligible to be part of the study. These could be seen as retrospective inclusion criteria, or framed to be retrospective exclusion criteria, but they do not determine someone’s eligibility prior to data collection.

Line 100: Double “method” here.

Lines 104–106: I’m uncomfortable with “These constructs were…” including “other”; could it be made clear here that this is not a construct, but rather an option for participants to select? You do this later on Lines 113–114, but it would be good to get this point clear slightly sooner, I think. Related to this, the definition on Lines 132–133 doesn’t seem meaningful to me as it would change its meaning if you added or removed a construct.

Line 123: What particular form of bias?

Lines 130–131: The 4th definition is just a rewording of the term. Could you clarify the meaning more, perhaps by explaining what “distress” means? This is a point also raised by Reviewer #2.

Line 137: “an empirically-based [focused?] publication”?

Line 144: This is very pedantic, but both of these dates could be DMY or MDY.

Line 144: “Of” rather than “From”?

Lines 145–149: I’m not sure why the list items start with a capital letter, and there is no “and” before the final item.

Line 146: “instrument” (not plural)?

Line 149: “…no English version was available…”?

Line 158: It might be useful to (briefly) discuss overlap between these two items and other items in the study. For example, the second omitted item is not completely unrelated to the example item for the PCS on Lines 163–164 and the example item for the CSQ on Lines 167–168.

Line 193: Does the reader need to know the values in this coding scheme? I can see how it helps explain the final score calculation, but this could also be done by explaining that for negative responses to the first question, the confidence score was turned into a negative value. I’m not convinced that this would necessarily be clearer than the current version, but it’s worth considering.

Line 201: Can you just confirm that this is gender, the social construct, and not sex you asked about?

Line 201: The first language was presumably to check the inclusion criteria?

Lines 201–202: It would be very helpful to list the levels for ethnicity, marital status, profession, and education here.

Line 205: I think there is a missing closing parenthesis here.

Lines 205–209: Because one list item includes commas, this would be clearer with list items separated by semi-colons.

Line 210: Are these actually Likert items or just ordinal ones?

Line 211: “pain-related”.

Lines 214–215: I think that some readers would appreciate measures of reliability and validity being included here.

Line 230: You might consider deleting “Indeed” here.

Line 230: I am assuming this assignment was randomised? Was this simple or stratified randomisation if this is the case?

Line 241: The second decimal place for the mean minutes seems overly precise (and inconsistent with the decimal places for the standard deviation).

Lines 244–245: Please give versions of packages used and for R to assist readers with replication.

Line 247: The wording in this description of random effects isn’t quite correct (the deviations are for effects not factors). If you have the second edition, see pages 13 and 14 (sections 2.1.2.3 and 2.1.2.4) of West, Welch, and Galecki.

Lines 248–252: This doesn’t seem to be a complete sentence (see also Review #1’s comments).

Line 252: Do you mean that no reverse coded items were selected for consideration, that they were reverse coded before analysis, or that they were specifically excluded from analysis because of being reverse coded?

Line 253: Was order of presentation included/considered as a fixed effect? Given the possibility of fatigue, I would have thought that systematic trending upwards or downwards was possible and worth modelling.

Lines 253–255: It’s not clear to me why you would ever even consider a model that did not accommodate known clustering. Can you justify this step? More information about how the final models were determined is needed here.

Lines 255–256: This might be stated more simply that a random participant effect was used, along with details of nested random effects within participant (based on Line 297).

Lines 256–257: REML is an estimation approach; it will not in itself determine the “best pattern for the covariance”. Were specific residual covariance structures examined, as you only mention random effects? Given a single random effect, you are (effectively) looking at compound symmetry only for participants.

Lines 257–258: See comment about Lines 253–255 above.

Lines 258–260: Can you make it clearer what variables were considered in the methods here? This should also include any interactions.

Lines 260–261: You’ve said above that you used REML, which is a different estimation method to standard ML, and one that has desirable properties in general. Can you explain more precisely what you mean here? Given you are selecting models based on AIC, the issues with using REML when selecting a model (REML precluding likelihood ratio tests) are not important here. If ML was used, a likelihood ratio test would be more usual, but ML would be less desirable for estimation purposes. Can you justify the procedure used here? As a note, if parsimony was your goal, BIC would have been a more usual choice.

Line 265: There did not appear to be any discussion of model diagnostics in this section. This is important and your chosen diagnostics need to be presented here so the reader can judge whether they would expect the resulting models to be satisfactory. Note that at least one of the 95% CIs in Table 1 goes outside the possible range for the data (-10 to 10) and at least three more come close enough to cause some concerns about the estimation. Mixed truncated regression would be one option here, or you could possibly argue that the scores are censored and use mixed tobit regression. In the Supplementary Table, there are at least seven more 95% CIs that extend outside the possible set of responses (-10 to 10).

Line 265: While clear enough here, and you note this for Table 1, it is traditional to include the level of significance used in the statistical methods. Given the number of tests performed here, it would be worth including a short explanation for, I am assuming, not adjusting for multiplicity.

Line 269: Perhaps “…data on…” rather than “…data of…”.

Line 272: These exclusions based on first language surprised me. Was this not included in the participant information and asked prior to data collection?

Line 281: Please specifically indicate the 2.22 as a mean (assuming this is the case). Note that for that mean and the reported SD, there is clearly positive skew and this calls into question the mean and SD as appropriate summary statistics. Medians and IQRs, or geometric means and geometric SDs for strictly positive measures, might have been more appropriate. Similar issues (both making it clear what each number represents, which is done consistently for SDs but not for apparent means, and skew) apply for Lines 281–286. For Lines 283, 284, 285, and 286, “average” is a vague term (and could refer to arithmetic mean, median, mode or some other indicator of location).

Line 287: Perhaps either “…that only a few participants…” or “…that few participants…” here.

Line 292: “received” not “receiving”.

Lines 292–303: It does not make sense to statistically compare the three groups if they were randomly assigned unless you are specifically looking for something that did differ at or due to randomisation, which I think here would be limited to priming effects from the other items. See the ample literature on comparing baseline characteristics between randomised groups in RCTs (where statistical significance would indicate that differences were unlikely to be due to chance, which leads either to concluding that these are in fact chance differences or that they provide evidence that the randomisation process was compromised—neither of which is a useful interpretation in most cases). If you feel that there is a valid question being addressed here, please provide a justification for these analyses in your rebuttal. The same issue arises for Lines 304–314. In order for these analyses to be valid, I think you would need to argue that the item sets might affect the response to these questions, which for gender and age seems utterly implausible and for pain status seems unlikely to me (and would require this to be asked after the item set to even be possible). Note that these tests should be described in the statistical methods, along with appropriate descriptions of model diagnostics performed.

Line 297: I think the choice of measure as a random effect here is worth some discussion in the manuscript so the reader can understand why you choose to model this as a random and not a fixed effect. There are theoretical aspects to this decision and the interested reader will want to appreciate your point of view here. The same point arises with Line 322 in general, item on Lines 324–325, etc.

Line 312: “In summary…”. Also Lines 366,387, 460, 479, 498, and 579.

Lines 320–322: I think it would be helpful for the reader if the information for each set of models, including those described here, was included in the statistical analysis section. You could still remind the reader here and indicate which effects were included in the final model, but having all of the model details together would be helpful, I think. This would help to address the related comment from Reviewer #1.

Line 322: I’m concerned that the nature of some variables, item for example, varies between each of these “check” models (whether they are included as a random effect or not), making the interpretation of the models incompatible in a particular sense. It appears to me that random effects are being included based on model fitting, which I consider problematic due to this change in interpretation, along with the usual caveats about model selection, and so even for models within sets of models, the random part of the model specification varies. I also feel that some of these models discussed later have tried to include more random effects than reasonable (perhaps explaining Lines 257–258). Can the authors justify their approach to model selection and make the exact process they used clear in the statistical analysis section? Also applies to Lines 403–404. See also the comments from Reviewer #1 mentioned above.

Lines 323–365: There is a lot of material here and given that this is to check the associations between items and their expected construct, I wonder if there are other ways to either analyse this (a confirmatory factor analysis perhaps—this would match the form of your summary on Lines 366–367 perfectly) or to present this (a forest plot with items as rows, organised by instrument, or with mean item scores for each instrument might just about work here, particularly if you use a letter system to show differences between constructs—you could also make pain catastrophizing the reference construct in the plot as it is of key interest and show differences for other constructs, or show only the construct with the highest mean, or some other approach). The approach used in Table 1 or Figures 1 and 2 could also be used. I think if you could find a graphical (or even a tabular) presentation, the reader would follow this section much more easily and much faster.

Line 326: The symbol mu normally refers to the (unknown) population mean. I’m assuming these means were calculated using BLUPs? Again, there are details that some readers will want to have made clear. The same symbol is used elsewhere.

Line 334: This suggests that the slope for construct is a random coefficient, which doesn’t make sense to me as construct is not a continuous variable. Am I misunderstanding here, with another variable being the slope? Can you make this part clearer? Also on Lines 350, 359, 407, 416, 425, 443, and possibly elsewhere.

Lines 361–362: The order of reporting differs here (normally mean, CI, t-statistic, p-value).

Line 400: “fit”.

Line 487: I suggest a comma after “analyses”.

Lines 491–494: I think readers will be unsatisfied to see that there is, for example, a construct by gender interaction without knowing more about how construct effects differed between males and females (even if you feel the interpretation to be similar). As queried by Reviewer #1, I would prefer the additional results mentioned in the manuscript to be included as supplementary material rather than available on request as I think these will be of interest to a reasonable fraction of potential readers.

Line 499: While I appreciate the point that I think you are making around effect modification not meaningfully changing results, the presence of these statistically significant interactions does contradict one reading of Line 499 as it is currently written.

Line 505: I think you need to be very careful about this and similar statements below. Your results depend on participant ratings and while we hope that these reflect the truth, there are threats to this, including external validity (e.g. Lines 606–608), the data collection modality, etc. If you agree, this could be as simple as adding “appear to” before “measure” here, with a related edit on Line 507 (“appearing to measure”). You seem to reach the same conclusions around Lines 566–568, so perhaps some foreshadowing here would be useful.

Lines 517–518: This is more important than it might appear to the reader at this stage. Given the study design, you have not been able to check that no items were missing; relying instead on all possible items (excluding variants) being included in those existing instruments included here. To me, this this the crucial challenge when moving from the initial face validity stage to content validity in its broader sense (see Reviewer #1’s comments on this point also). I don’t think it’s at all likely that highly important aspects of pain catastrophizing are missing from the existing instruments, but this is a limitation and one that did not appear to be covered by the limitations later (Lines 606–626), unless you wish to restrict this to discriminant content validity (and do the same consistently throughout the manuscript).

Line 525: “These guidelines provide guidance” is perhaps tautological and could be simplified by omitting “guidelines” here.

Lines 547–548: This might be reflecting my conservative nature, but I think a “This appeared not to be the case.” here would make it clear that this is an empirical observation only.

Line 550: Perhaps “offered” rather than “did offer”.

Line 553: You could also add “as understood by our participants” here at the end of this clause.

Lines 560–561: I think that a note that there was, however, evidence for effect modification would be useful here to remind the reader that your argument is not, as I understand it, that there is no evidence of differences but rather that any such differences are not of practical importance.

Lines 561–562: Sorry for the repetition, but again I think you need to soften this point. There was no evidence of such differences does not support an absolute statement that there are no differences, and in any case there were some statistically significant interactions involving pain group (Lines 491–494).

Lines 566–587: This is just an observation, but it is interesting to me that the key finding here (along with the overlap between constructs as currently measured) is one that could have been obtained a priori. This line of argument does not, despite the start of the first sentence immediately before, hinge on your results and could stand alone from any empirical study, which might explain to some extent the length of your manuscript.

Lines 621–628: While I think this is a useful point to note, it would not be a limitation of your research (a reason to suspect the findings) for me.

Line 628: I think the sample size and widths of the 95% CIs needs to be included either as a limitation (the realised power of the study as indicated by the 95% CIs being too wide) or a strength (the widths being sufficiently narrow). The strengths could remain where they are (Lines 554–562), or you might wish to move them alongside the limitations here.

Line 632: “…will ever be.”?

I think you need another table describing the participants based on their characteristics (Lines 200–203). This could perhaps be overall and stratified by pain group.

Table 1: There is a missing space in “scoringpain”. The symbol mu is used to refer to population means (and is “mu” not “mhu” as written here) and the x-bar symbol would be the usual for the sample mean when used to estimate the population mean (rather than mu with a caret indicating that it is an estimate—note that the caret did not seem to appear in the text). The same point applies to the Supplementary Table (and note also the missing space after “CATASTROPHIZING”).

Figures 1–2: These should not be line graphs (as also noted by Reviewer #1).

For Figure 1, unless I’m missing something, the asterisk is unnecessary as any 95% CI (I’m assuming that this is what is shown and this should be made explicit) that overlaps with zero will not be statistically significantly different from zero. I appreciate that if there were closer cases the asterisk might become important, but it seems to be that its omission here wouldn’t hinder the reader from interpreting the graph. This is entirely up to you though.

For Figure 2, there is a missing space in “Eachfigure”. Again the meaning of the error bars should be made clear (in other fields, these could be standard error bars, for example).

·

Basic reporting

a few small points :

line 211 – ‘participants’ not ‘patients’
line 250 lacks a verb e.g. ‘amongst which were the constructs…’
line 263 should be ‘by’ not ‘with’.
line 495 ‘from’ not ‘as’.
line 632 ‘It is unlikely that it will ever be.’

line 496-7 Is it appropriate to have ‘available on request’ rather than as supplementary?

Experimental design

no comment

Validity of the findings

no comment

Additional comments

Review of Crombez et al: Lets talk about pain catastrophizing measures: an item content analysis.

This paper reports a study in which 94 on-line participants rated the relevance of 58 items from six standard pain catastrophizing measures to six pain constructs, including pain catastrophizing. They found that two measures did not measure pain catastrophizing and that none of the measures pain catastrophizing alone. Most of the questionnaires measured either pain worrying or pain distress more than they measured pain catastrophizing. This is an important paper and I strongly support its publication.

The strong points of this paper are:
1. It addresses an important question. Pain is a central scientific and practical domain and the results of this study are important scientifically and practically. Scientifically, the proliferation of constructs that cannot be distinguished operationally is of little value and may be damaging (Dixon & Johnston, 2019). If measures are contaminated by other constructs, tests of theory are invalid and may lead to the design of inappropriate interventions and experimental conditions. Practically, decisions may be made and services planned on the basis of misleading information: as the authors point out, patients scoring high on pain catastrophizing may be referred to mental health services whereas this would not occur if the measure was more correctly labelled ‘pain distress’ or ‘pain worry’.
2. The findings are clear and important for methods and theory in this field.
3. The methods are excellent. They advance on our earlier work (Johnston et al., 2014) by the use of more elaborated statistics. In addition the manipulations checks and tests for the level of engagement of participants in the task are excellent.
4. The Introduction and Discussion present the issues clearly.
5. The Results, including the rationale for decisions that had to be made, are presented very systematically and clearly.
6. It is very useful to have both the overall analysis and the analysis for each questionnaire. In particular, the PCL measure looks extremely poor.

Some queries
1. The definitions are crucial in this kind of study and merit some discussion. The authors offer a definition of pain catastrophizing as ‘to view the present pain of pain-related problems as considerably worse than they are.’ I do not work on the pain catastrophizing so cannot comment on whether that is representative of definitions in that area, but other definitions of ‘catastrophising’ would not require the deviation from the ‘actual’, e.g. defining it as assuming or anticipating the worst. Given that it is difficult to say what pain ‘actually’ is, might these definitions be more useful? Second, the study used dictionary definitions for other constructs and, as shown in Bell et al (2017), this may not fully represent the construct as used scientifically. Third, the definition should appear earlier in the Abstract – in either Background or Method rather than Conclusion – and earlier in the Introduction to the paper.
2. The difference between content validity and discriminant content validity gets blurred e.g. lines 84-90 could make the separation clearer. The Abstract Background says ‘we investigated the content validity of self-report measures of pain catastrophising’. This omits to say that a key part of this paper is detecting whether these measures have discriminant content validity that distinguish them from other pain measures. The contrast between these two questions is sometimes not clear at various points in the main paper.
3. It would be useful to present a discussion of the choice of statistical analysis – including the pros and cons compared with Johnston et al 2014 Methods.
4. If possible, a discussion of the number of participants required would be useful – and might be important in discussing a comparison of the requirements of this method compared with Johnston et al 2014.
5. Line 64-72. The argument here could be argued more strongly and clearly. In particular the first point is barely made – why should the pain catastrophizing measures correlate with the CEQ?
6. Should the Figures be histograms rather than graphs as the order of items and intervening points have no meaning?
7. In the Discussion in lines 568 onwards, the challenge of measurement of pain catastrophising depends on the definition and it would be useful to discuss this – see point 1 above.

Small points
1. It would be useful to indicate in section 2.2.3 where the full list of items was available.
2. Line 211 – ‘participants’ not ‘patients’
3. Line 204 – is it useful to mention that the CPG has discriminant content validity (Dixon, Pollard & Johnston, 2007)?
4. Line 250 lacks a verb e.g. ‘amongst which were the constructs…’
5. Line 263 should be ‘by’ not ‘with’.
6. Line 495 ‘from’ not ‘as’.
7. Line 496-7 Is it appropriate to have ‘available on request’ rather than as supplementary?
8. Line 632 ‘It is unlikely that it will ever be.’

Bell, C., Johnston, D., Allan, J., Pollard, B., & Johnston, M. (2017). What do Demand‐Control and Effort‐Reward work stress questionnaires really measure? A discriminant content validity study of relevance and representativeness of measures. British journal of health psychology, 22(2), 295-329.
Dixon, D & Johnston, M (2019) British Journal of Health Psychology (2019) © 2019 The British Psychological Society www.wileyonlinelibrary.com Editorial Content validity of measures of theoretical constructs in health psychology: Discriminant content validity is needed. BJ Health Psychology. https://rdcu.be/bA14H
Dixon, D., Pollard, B., & Johnston, M. (2007). What does the chronic pain grade questionnaire measure? Pain, 130(3), 249–253. https://doi.org/10.1016/j.pain.2006.12.004

·

Basic reporting

This article is well-written and well-structured, meeting the highest standards of basic reporting throughout. The authors use clear, professional English which makes the study and their discussion of results easy to follow. The authors provide adequate consideration of the previous literature, including appropriate references. The authors clearly state their results and provide useful tables.

I have only minor suggestions to improve the basic reporting of this well-written paper:

1) Can the authors provide the exact scoring they used to calculate Pain GRADE (i.e., what constitutes low/high intensity and disability)? Although this approach stems from the original Von Korff et al paper, the extant literature has used slightly different calculations and approaches. In particular, while the authors chose to have Grade 0 reflect ‘pain free’ participants, other studies have had this grade reflect ‘no chronic pain’ (i.e., pain < 3 months is included).
2) Can the authors provide a brief description of the frequency of use of the selected pain catastrophizing measures? My understanding is that the PCS is by far the most commonly used, and some of the measures have not been used for a number of years. This summary would be useful to situate the results within an understanding of how the selected catastrophizing measures are typically used in the field today.

Experimental design

The manuscript presents novel, original work that is both within the aims and scope of PeerJ and will be of significant interest to the field of pain science. The research question - whether pain catastrophizing measures indeed measure pain-related catastrophizing, as defined by cognitive-affective theory, is clear and important. The work is timely and fills an identified knowledge gap.

The experimental design is largely appropriate. However, there are two significant methodological issues (detailed below) that are problematic for the conclusions drawn by the authors, and which should be addressed in the manuscript before publication.

1) I find the authors' conceptualisation of 'pain-related distress' problematic. This is important given that one of the primary conclusions of the study is that pain catastrophizing measures are better represented by pain-related distress (and pain-related worry). This is likely to be a well-cited and repeated finding that drives future research in this area. Specifically, unlike concepts such as worry, rumination, and vigilance, the concept of 'distress' is not well-defined in either cognitive-affective theory or in pain science more broadly. The term 'distress' is often used as an umbrella term to capture elevated arousal states including fear, anxiety, catastrophizing, and (hyper)vigilance, both in the affective science literature and in pain science. Partly in reflection of this, the definition that the authors provide for pain-related distress ("Distress related to pain or pain-related problems") is insufficient. The fact that the definition includes the very word that it is trying to define exposes this problem. Relatedly, the measure that the authors chose to represent pain-related distress was actually an affective distress measure, with no relation to pain (i.e., "in the past week, how tense or anxious have you been"). These issues have implications for the authors' work in a few ways. First, given that the term 'distress' is used in a catch-all way, and that the definition provided to participants is so general, is it perhaps unsurprising that the participants rated catastrophizing items as also reflecting the more general construct of distress. Further evidence for this is the fact that worry and distress items were not rated as different, either. Second, readers of this paper will likely start using the term 'pain-related distress' in place of catastrophizing in their own work, believing it is an adequate (i.e., well-defined) replacement of pain catastrophizing. Given the issues described above, I don't believe this is the case. Overall, this issue is perhaps why the authors chose to recommend use of the term ‘pain-related worry’ instead of distress, in place of catastrophizing. But I believe this issue needs to be better addressed throughout the manuscript, including explicit discussion of why the current findings are not necessarily sufficient to suggest using the term pain-related distress are a replacement term as of yet.


2) Another major conclusion from the study is that pain catastrophizing measures are not well-defined/specified *in comparison* to the other constructs considered in this paper (i.e., pain severity, pain-related worrying, pain-related vigilance, pain-related distress and pain-related disability). In the abstract the authors state: “The results revealed that items from pain severity, pain-related worrying, vigilance, distress and disability scales were distinctively related to their respective construct. This was not the case for the items from the pain catastrophizing measures.” The issue with this conclusion is that the other constructs were not submitted to the same procedure as the pain catastrophizing measures. Specifically, for the other constructs the authors selected only one measure that was ‘considered appropriate for the respective construct’. Thus, they selected the single most appropriate measure these constructs, while for pain catastrophizing they selected a number of measures, many of which were developed decades ago and have not been in mainstream use for a while. The authors’ additional analyses showing that even the ‘best’ 5 catastrophizing items did not diverge from measures of worry/distress points toward is useful for the authors’ conclusion, but I feel that overall it is overstated. This is especially the case given that pain-related worrying items were not distinct from pain-related distress, indicating issues with even the ‘most appropriate’ single measures of these constructs.

Validity of the findings

There are a number of strengths of the authors' methodological approach that leads to robust, valid findings. For example, they used methods to detect careless responding, which is important for an online sample. They also confirmed that there were no differences in the way participants reporting persistent pain rated items, in comparison to those without pain issues. I have only one issue with the conclusion that the authors draw from their work, described below.


The authors make a strong recommendation at the end of the manuscript – to reclassify pain catastrophizing measures as measures of pain-related worry. The results of this study, ie., that pain catastrophizing items were often rated as reflecting pain-related worry, provides compelling evidence to support his suggestion. However, there are other considerations that I believe need to be addressed, as caveats. The concept of ‘worry’ is very well defined in the cognitive-affective literature, particularly in relation to work on anxiety disorders, and the available pain catastrophizing measures do not necessarily reflect the complexity of ‘worry’ as it is understood more broadly. For example, worry is typically understood as a process of repetitive negative thinking about an event in the future. This is distinct from rumination, which is understood as a process of negative repetitive thinking about events in the past. The available pain catastrophizing measures, including the PCS (which is the most widely used), do not typically distinguish between these. An alternative recommendation that the authors could have made, particularly pertinent for the PCS, is to consider and utilize the constituent subscales instead. Specifically, the PCS has three subscales: rumination, magnification, and helplessness. I was surprised that the authors did not consider including these constructs in their study design – specifically examining how participants would rate the subscale items as reflecting their respective subscale definitions. This data would have been particularly useful for considering the alternative approach of focusing on subscales. Can the authors address this issue and include some consideration to this as an alternative way forward (including the necessary studies for examining the utility of this approach)?

---

## Round 0.2 · Minor Revisions

Thank you very much for the revised version of your manuscript. The reviewer for this round has indicated that they are happy with the manuscript as it is (a suggestion on their part notwithstanding). I’ve made a number of comments below, many of which are editing suggestions, which, should the manuscript be accepted, will help make the proofing stage much simpler. Where my suggestions are stylistic, please do feel entirely free to reject them, but also, please do make sure that you are consistent throughout the manuscript in terms of small numbers in numerals versus words, the use of Oxford commas, etc.

I have also raised a small number of slightly more important questions around the new models and their reporting, but I don’t think these will be too challenging for you to respond to.

I couldn’t find the second supplementary information file (S2, Line 422) although I did see the supplementary table S1. My apologies if I am overlooking something as a few of my questions below would be answered in that supplement.

I think you’ve done a wonderful job of re-analysing the data and presenting the results with admirable clarity. I’m sure that you will be able to address my comments below without too much difficulty and I think that this manuscript would be an invaluable addition to the pain literature at that point.

Line 37: I suggest a comma after “Based upon current findings”.

Line 38: I suggest removing the comma in “we argue that pain catastrophizing, cannot be assessed”.

Line 81: You could give a more specific link here, e.g. https://www.oed.com/view/Entry/28799. The phrase “as considerably worse than it actually is” is verbatim from the definitions there and so should be quoted if it originated from there (you quote a slightly longer and slightly different definition on Lines 457–458, Lines 494–495, and Lines 570–571)

Lines 90 and 91: Should the “i.e.” (in other words) on each line be “e.g.” (for example) as presumably these are not exhaustive lists? See also Lines 153, 154, and 156, and perhaps elsewhere also.

Line 98: Perhaps an “and” before “(3)” as this is the final item in the list. You do this sometimes (e.g. Lines 218 and 278), but not always (some exceptions are noted below) and you should check the manuscript to make this consistent.

Line 103: Is the uppercase “D” here warranted (if so, should C and V also be capitalised)? Also, it seems odd to provide the abbreviated form in the heading above, particularly when it is not then used on the following line, and so this might be better done on this line (i.e., by moving “(DCV)” from there to here).

Line 113: I suggest removing the comma in “The categories, ‘pain severity’”.

Lines 117–118: I don’t think you need a hyphen in “‘other’-category” (note that you don’t use one elsewhere, e.g. Lines 119, 122–123 and 123). See also Line 139 where this hyphen appears again.

Line 126: I suggest adding a comma after “For each of the identified constructs”.

Line 138: Perhaps an “and” before “(6)” as this is the final item in the list.

Line 144: Perhaps an “and” before “(3)” as this is the final item in the list.

Lines 152–158: I think that these sentences would work better as list items (1) through (4).

Line 165: The “1” in “10 items1,” should be a superscript.

Lines 178–179: I think you have a missing word here: “…of which 3 WERE reverse coded items…” Note that you use “five” (as I would use) on the previous line, so perhaps change “3” to “three” here for consistency. The use of numerals or words for small numbers is inconsistent throughout the manuscript (e.g. Line’s 192 “three items” and Line 196’s “3 items”) and it would be helpful to adopt a single approach for the entire work by checking the manuscript.

Line 179: Is “Thinking too much about the pain only makes it worse” a reverse scored item (higher values would presumably indicate greater catastrophizing)? I couldn’t find this item in Table S1. Your example on Lines 196–197 is “I find it easy to ignore pain”, which is clearly reversed.

Line 186: I think “retrieved of” here should be “retrieved FROM” (as you use on Lines 187–188, Line 190, Lines 192–193, and Line 195 just below).

Line 218: Rather than “and the (7) number”, I think “and (7) the number” would read more naturally.

Line 233: I’d have a comma before “in their honest opinion” as well as after.

Line 238: Perhaps “link TO the survey” rather than “of”.

Line 242: I’d suggest “was split inTO three item sets” here.

Line 244: It would be helpful to readers if you clarified what the strata variable was in “stratified randomization”.

Line 251: Due to the potentially ambiguous nature of “average” (could be arithmetic mean, median, mode, etc.), perhaps say “lasted A MEAN OF 35.4” here. Also Lines 300, 301, 302, 303, 305, 306, 307, and perhaps elsewhere.

Line 258: Could you briefly identify the “vague priors” used (e.g. Jeffreys, normal with a very large standard deviation, uniform, or something else). This is important, at least to a subset of your potential readers, as some seemingly vague priors can have pathological effects in some situations. Perhaps I was looking in the wrong place, or perhaps this has not been updated yet, but I couldn’t see the code for the JAGS models on OSF to check this myself.

Line 264: Rather than “difficulties to score the reverse coded items”, perhaps “difficulties IN SCORING the reverse coded items”.

Line 269: I suggest “as A fixed effect” (adding “a”). Also Lines 273, 279, and 281 (for the last of these, both “fixed effect” and “random effect” would have the article added.)

Lines 270–273: If the purpose of this analysis is to investigate the measures, these are rightly included as fixed effects (you’re not trying to generalise beyond this set). However, this would then make item a (nested) fixed effect for me as each measure comprises a fixed set of items (you’re not treating the items in the AEQ, for example, as a random sample of possible items from the AEQ, this is the full set of items from that instrument of interest here). Can you please consider this point and explain your approach here? For other analyses, the items used seem indeed to be a random effect as they are a sample from the items in the used and other (including not yet developed) instruments.

Line 271: I suggest “and” before “PRSS” here as it’s the last item in the list.

Line 274: I suggest “were” rather than “was” here.

Lines 282: I think that this would be a good place to foreshadow effect modification/interactions by listing the models investigated here. The models used to produce the results on Lines 419–422 do not seem to be discussed in the methods as far as I can tell. If group was included in interactions (Line 415), this would also apply. If the examination of effect modification was purely descriptive (supplement S2 did not appear to be included, which would presumably clarify this; my apologies if I’m overlooking this supplement somehow), this should also be made clear here as some of the descriptions in the results (e.g. Lines 419–421) and the discussion (e.g. Lines 486–487 and Lines 488–489) lead me to expect that interactions were examined to formally test for effect modification. The comparisons of baseline characteristics between groups should, as discussed previously, be descriptive only, but again this will be in supplement S2 and I was unable to check this.

Line 282: Presumably this is two-sided 5% and this should be made explicit.

Lines 289–292: These semi-colons could be replaced with commas as none of the list items include commas.

Line 293: You could provide the reader with the percentages as well as the counts (to save some of them reaching for their calculator) here and on Lines 294–297, Line 298, and Lines 308–310. For Line 293, you could also use “n = 59” for consistency with the following lines, but this is definitely stylistic.

Line 298: I suggest “were” rather than “was” here.

Line 306: I suggest deleting the “to be” in “Finally, participants reported to be on”.

Line 312: I suggest adding a comma after “As a manipulation check”.

Line 347: I suggest “were” rather than “was” here.

Line 348: I suggest a comma after “For each catastrophizing questionnaire”.

Line 355: This is a very minor point, but here you don’t have a comma before the “and” in the list; but you do use this (serial/Oxford/Harvard) comma in other places, such as Lines 347 and 350. It would be worth checking that you are consistently applying your rule for this comma.

Line 397: I suggest “scoreS” here.

Line 401: I suggest “mean” rather than “average” here, or “median” if that was intended instead.

Line 409: I suggest a comma between “better” and “captured” here.

Line 480: I suggest a comma before (as well as the one after) “as understood by our participants”.

Line 486: You use “worringy” here.

Line 504: This is merely a suggestion, but given emerging attitudes towards gender pronouns, you could consider “they” rather than “(s)he” here to avoid the dichotomy.

Line 541: “one-tailed” would refer to the level of significance (only rejecting the null if the test statistic was significant in a pre-specified direction) whereas “paired t-tests” would be for matched data and so looking at differences. I’m wondering if there are two separate issues here? (Johnston, et al. in the first paragraph of their results talk about both paired t-tests and one-tailed t-tests, but the latter is concerned with the level of significance rather than the test per se and one-tailed tests should only be used when, a priori, differences in the other direction would not be of interest.) You could say “one-tailed paired t-tests” here to capture both aspects, although they do also talk about signed rank tests as an alternative (for when the differences are not normally distributed, which for their smaller sample sizes is harder to establish empirically).

Line 544: Do you mean “Notwithstanding THIS, it…”?

Lines 544–545: The items in this list don’t themselves contain commas and so you could downgrade the semi-colons to commas. (And there is no “and” here before the last list item.)

Line 556: “we did only address the content validity” might be clearer as “we only addressED the content validity”.

Line 560: While you say “Seventh”, you also used this on Line 556, so this would be “Eighth”.

Line 564: You talk about extreme “judgment” here, but you talked about extreme “perception” back on Line 503. Is it worth being consistent between these or am I missing some distinction here? “perception” seems to better match with the “view” that you have in your standard definition and I’m not sure that “judgement” matches the “present” aspect of that definition.

Line 565: You’ll want a comma after “If assessed through decontextualized self-report”.

Line 572: Rather than “It is unlikely that it will ever do.”, I suggest “It is unlikely that it ever will.”

Table 1: I’m assuming the “EMM” under “Pain-related disability” should also be italicised? Same for “Pain vigilance”. Note that the legend says that “mu = estimated mean in the population” but the table is headed with “EMM”s instead.

Table 1: I’m assuming the negative sign in “6.91* [-5.81 to 8.04]” is spurious (vigilance and protect myself).

Figure 1: I don’t think you need the comma in “The estimate and CI for the construct each measure is designed to assess, is indicated in black.”

Figure 1: Spurious spaces after hyphens in “(C) MPI- affective distress”, “(D) MPI- pain-related disability”, and “(E) MPI- pain severity” (colons could work in this context also). Note that in the figure itself, you seem to have both leading and trailing spaces around the hyphens, which would also work as an em-dash.

Figure 3: I don’t think you need the comma in “The estimate and CI for the construct each measure is designed to assess, is indicated in black.”

Lines 596–597: Source title is not italicised.

Line 626: I’m not sure why PAIN is in all capitals here (c.f. Line 631). See also Line 752, 754, 780, and 789.

Lines 636–637: Source title is not italicised.

Line 648: Source title is not italicised.

Lines 668–669: Source title is not italicised.

Line 796: Source title is not italicised.

·

Basic reporting

The manuscript remains very well written and I appreciate the authors' need for brevity in not adding too much additional information.

Experimental design

The authors have refocused their suggested terminology to pain-related worry, and have provided more caveated discussion about the pain-related distress term. I appreciate that the authors revised some of their language to ensure that they were not making the claim that the other constructs were better defined that pain catastrophizing.

Validity of the findings

Although I still think that the authors could make an explicit statement about the use of the PCS subscales (rather than talking more broadly about 'constituent components'), I believe that the findings of the paper are valid and robust and meet high standards.

Additional comments

The authors have clearly taken a great deal of time to respond to all of the reviewer and editor's comments. I believe this paper will make an important contribution to the field.

---

## Round 0.3 · accepted · Accept

Thank you for your revisions and responses. I am delighted to accept your manuscript. I will point out a few small inconsistencies and typos below, but these can be corrected, as you see fit for the stylistic points, in the proofing stage rather than requiring another round of revisions. Well done on writing a thought-provoking manuscript and I look forward to the discussion that I am sure it will generate.

Lines 27–28 and 30–31: This is a very pedantic point, but the list items are the same in both places but are listed in a different order. The second order is used just below on Lines 33–34.

Lines 27–28: There are still some inconsistencies in the use of Oxford commas—not used here or on Lines 31, 33, etc. but used on Lines 46, 63, 91, etc.

Line 66: There is a spurious “s” either after “doubt” or after “exist” here in “First, doubts exists whether…”

Line 157: The three preceding list items all begin with a capital letter (“No”, “The”, and “No”), but a lower case “a” is used here “(4) a standard item format…”.

Line 193: The opening single quote here is reversed (i.e. a closing quote): “For ’pain-related disability’,”. This also arises twice on Line 221 and on Line 411 (and possibly elsewhere).

Line 220: Normally, outside of parenthetical references, you use “and colleagues” rather than “et al.” as you do use here. (C.f. Lines 48, 50, 199, 462, 464, and 479.)

Line 275: There is a missing “s” after “effect” here. (C.f. Lines 270 and 282.)

Line 297: For consistency, and as a kindness to readers, you could add the percentage of males to the count here (as you do for ethnicity on Line 298, relationship status on Line 299, etc.).

Line 311: It might be useful to remind readers that this number of days is over a six-month period here.

Line 318: “pain-related disability” is not quoted here but the other constructs on Lines 317–318 are all quoted.

Line 331: Did you want an “and” before the final item (“PRSS”) here? (C.f. Line 337.)

Line 334: Is the use of “‘catastrophizing’” rather than “‘pain catastrophizing’” intentional here? (C.f. Lines 330, 332, and 333, as examples.)

Lines 338, 339, 340–341, 342, and 343: It seems that “pain catastrophizing” is briefly unquoted here, including two explicit references to the construct). Are these all intentional?

Lines 376–377: Perhaps “There were no significant differences with the scores on…” (as there are two other constructs). A similar comment applies to Lines 389–391.

Line 406: I’d add a comma after “before” here (as you do on Lines 410–411) and remove the comma between the two list items (as you do on Line 411).

Lines 406 and 411: I notice that you use “‘worry about pain’” on the former and “‘worrying about pain’” on the latter. The first option seems to be only used here, Line 155, and in the legend for Figure 1, with the second option used in all other cases.

Line 423: The order here (“pain group, age, and gender”) differs from that a couple of lines above (Line 421: “age, gender and pain group”). This order also varies in the online supplement.

For the second supplement, the legend for Figure 4 could be improved (currently “GCPS_Pain” with values of 0 and 1).

Line 435: “measure” here should, I think, be “measuring”.

Line 436: I don’t think you need the comma in “measure pain catastrophizing, distinctively”.

Line 458: You capitalise “Oxford Living Dictionaries” on Line 131, but here you do not capitalise the last word in “Oxford Living dictionary”. See also Line 548: “Oxford Living dictionary”.

Line 486: You’re missing the opening single quote for “vigilance” here.

Line 490: I don’t think you need the comma in “the pattern of results, remained the same”.

Line 547: “confidence intervals” here should, I think, be “credibility intervals”. See also the legend for Table 1.

Line 575: The first “will” in “It is unlikely that it will ever will.” seems spurious (although it could also be the second that is spurious).